# Enhanced Acetone Sensing Property of a Sacrificial Template Based on Cubic-Like MOF-5 Doped by Ni Nanoparticles

**DOI:** 10.3390/nano10020386

**Published:** 2020-02-22

**Authors:** Ning Zhang, Huijun Li, Zhouqing Xu, Rui Yuan, Yongkun Xu, Yanyu Cui

**Affiliations:** Department of Chemistry and Chemical Engineering, Henan Polytechnic University, Jiaozuo 454000, China

**Keywords:** NiO/ZnO, cubic-like, VOCs, acetone, gas sensor

## Abstract

Studying an acetone sensor with prominent sensitivity and selectivity is of great significance for the development of portable diabetes monitoring system. In this paper, cubic-like NiO/ZnO composites with different contents of Ni^2+^ were successfully synthesized by modifying MOF-5 with Ni^2+^-doped. The structure and morphology of the prepared composites were characterized by XRD, XPS, and SEM. The experimental results show that the NiO/ZnO composite showed an enhanced gas sensing property to acetone compared to pure ZnO, and the composites showed the maximum response value when Ni^2+^ loading amount was 5 at%. The response value of the 5% NiO/ZnO composite to acetone (500 ppm) at the optimum operating temperature (340 °C) is 7.3 times as that of pure ZnO. At the same time, the 5% NiO/ZnO composite has excellent selectivity and reproducibility for acetone. The gas sensing mechanism of the heterojunction sensor was described.

## 1. Introduction

Volatile organic compounds (VOCs), as the important components of atmospheric pollutants, can easily exist in the environment as vapor at room temperature due to their high vapor pressure and easy evaporation [1,2,3]. Acetone is a toxic and highly volatile compound that tends to burn or explode when exposed to flame or high temperature. In addition, acetone is also an exhaled component of the human body, the concentrations of which can range from 0.3 to 4 ppm in the exhaled of healthy people. However, for an adult with diabetes, its concentration can rise to 1250 ppm [4,5,6]. Therefore, it is necessary to develop a portable gas acetone sensor for diabetes monitoring system to judge the blood glucose level according to the concentration of acetone in respiration [7].

Metal oxide gas sensors are extensively used in the detection and monitoring of various flammable, explosive, and toxic gases. However, the performance of metal oxide semiconductor (MOS) gas sensor used for the semiconductor materials is mainly affected by specific surface area, surface defects, morphology features, and adsorption capacity of semiconductor materials [8,9,10,11,12,13]. As a new type of organic-inorganic hybrid material, metal organic frameworks (MOFs) is a kind of well-organized crystal material with one-, two- or three-dimensional reticular framework self-assembled by metal ions and organic ligands [14,15]. A majority of MOFs has been reported that they have a wide range of applications in the fields of gas sensors, catalysis, adsorption and storage, quantum dot semiconductors. As a matter of fact, since the organic ligands in the materials can be decomposed into water and carbon dioxide by high temperature calcinations, the original porous and hollow structure could retain to a large extent in this way [16,17,18]. Moreover, MOFs have richer channels, higher specific surface area, adjustable pore structure, high porosity, and good stability compared with other materials, which not only make the MOF be used as an effective precursor or sacrificial template to fabricate sensitive materials for MOS, but also provide a new direction for the research of gas sensing materials [19,20,21,22,23].

However, there are many disadvantages in the gas sensing device prepared by ZnO obtained from a single MOF, such as low sensitivity, slow response, poor reliability etc., which cannot meet the requirements of industrial production. In order to improve the gas sensing properties of ZnO materials, doping with impurities is a simple and effective method to enhance gas sensing properties [24,25,26]. Nevertheless, due to the high cost of doping noble metals, it is difficult to apply to industrial mass production. Therefore, it has been widely concerned to doping other metal ions to enhance gas sensitivity performance. In this paper, the bimetallic MOF material was prepared by doping Ni^2+^ in the synthesis of MOF-5. Then, the Ni^2+^-doped MOF-5 was used as template to prepare bimetal semiconductor materials with improved sensitivity for acetone. The results showed that the product of 5% Ni@MOF-5 after calcining has the best gas sensing property toward acetone, and the response value to 500 ppm acetone is 97.134. Furthermore, we also discussed in detail the sensing mechanism of NiO/ZnO materials on acetone properties.

## 2. Materials and Methods

### 2.1. Materials

Zn(NO_3_)_2_·6H_2_O (99%), Ni(NO_3_)_2_·6H_2_O (98%), 1,4-dicarboxybenzene (H_2_DBC) and N, N-Dimethylformamide (DMF) (99.5%) were purchased from commercial suppliers. The required chemicals and reagents were utilized directly without further purification.

### 2.2. Synthesis of Samples

The NiO/ZnO structure was synthesized by an oil bath method. According to the literature reports, MOF-5 precursors were prepared and modified [27,28,29,30]. During the experiment, the mixture of Zn(NO_3_)_2_·6H_2_O (4 mmol), Ni(NO_3_)_2_·6H_2_O (x mmol), and H_2_DBC (2 mmol) in 50 mL DMF were sealed in a circular bottom flask of 150 mL and magnetically stirred at 130 °C for 8 h. The ratios of Ni^2+^ to Zn^2+^ were 0/1, 0.05/1, 0.25/1, and 0.5/1, respectively. The products were then collected with a centrifuge, washed three times by DMF to remove impurities, and dried in an oven at 60 °C. The obtained products were white powder (MOF-5) and light green powder (5% Ni@MOF-5, 25% Ni@MOF-5, and 50% Ni@MOF-5). Finally, the obtained products will be calcined at a heating rate of 5 °C min^−1^ for 2 h in a muffle furnace at 500 °C. For convenience, the ZnO structures doped with different contents of Ni were recorded as ZnO, 5% NiO/ZnO, 25% NiO/ZnO, and 50% NiO/ZnO, respectively.

### 2.3. Characterizations

The phase structures of the as-synthesized products were characterized by X-ray powder diffraction (XRD, Bruker/D8-Advance) with Cu Kα radiation at 40 kV and 25 mA over a 2θ range of 10–80°. The field emission scanning electron microscopy (FESEM, Quanta 250 FEG) with an acceleration voltage of 15.0 kV was used to observe the morphology of the samples. The energy dispersive spectrometer (EDS, Bruker) was integrated into the SEM system for elemental analysis. What is more, the chemical valences of the samples were studied by X-ray photoelectron spectroscopy (XPS, Thermo ESCALAB 250Xi) and Al Ka (1486.6 eV) monochromatic irradiation.

### 2.4. Gas Sensing Measurement of Samples

The preparation process of a gas sensor is described as follows. The prepared samples were mixed with deionized water in a mortar to form a homogenous paste and coated on a ceramic substrate screen-printed with Ag-Pd interdigitated electrodes (13.4 × 7 mm). Then, the prepared sensors were dried at 60 °C and aged at 240 °C for 24 h in order to improve the gas sensitivity of the samples.

The sensor test system and data acquisition system of CGS-4TPs (Beijing Elite Tech Co., Ltd., China) are used to test the gas sensitivity performance. The instrument converts the resistance of the sample to the display screen through electrical signals. In the experiment, the target gas was injected into the test chamber (1.8 L) with a microsyringe when the resistance of the material became stable at a specific temperature, and then the sensor was exposed to the target gas for 200 s to release the gas and return to the initial resistance. The experiment was carried out under the condition of relative humidity (RH) of 20% ± 2%. The response value is defined as the *R_a_/R_g_* ratio of acetone, where *R_a_* and *R_g_* were the resistance of the sensor exposed to air and acetone atmosphere, respectively. The response and recovery time were expressed as the time taken for the adsorption and desorption of acetone when the resistance of the sensor reaches saturation [26,31]. The following is the analysis scope and accuracy of the sensor test system and monitoring system. Analyzing range: 1 Ω~4 GΩ (precision: ±1 Ω, ±0.5% current value); temperature range: Room temperature ~500 °C (precision: 1 °C); time limit: >1 s, continuously adjustable; monitoring system: Chamber temperature (precision: 1 °C), and humidity (precision: 1%); evaporation: Room temperature ~250 °C (precision: 1 °C).

## 3. Results and Discussion

### 3.1. Sample Characterization

Appendix A displays the XRD pattern of MOF-5 with different Ni^2+^ contents. All composites (5% Ni@MOF-5, 25% Ni@MOF-5, and 50% Ni@MOF-5) have diffraction peaks as MOF-5, and no other diffraction peaks are found, showing that in situ doping of Ni^2+^ do not change the crystal structure of MOF-5. Appendix A indicates the XRD pattern of the oxide after calcining. The diffraction peaks of the sample ZnO are consistent with the PDF standard card of the hexagonal zincite structure ZnO (JCPDS: 36-1451, a = b = 3.250 Å, c = 5.027 Å). No other impurities of diffraction peaks are found, showing that the product obtained after calcining of MOF-5 is pure ZnO. No characteristic diffraction peaks of NiO of 5% NiO/ZnO are found in the XRD pattern, which may be due to the trace and high dispersion of Ni particles. In addition, significant peaks are observed at 2θ = 43.275° when the content of Ni particles is 25% and 50%, corresponding to (0 1 2) crystal plane of the cubic bunsenite structure (JCPDS: 47-1049, a = b = c = 4.177Å). No other diffraction peaks are found, showing that it contains no impurities and the sample is of high purity. In conclusion, we can conclude that Ni^2+^ has been successfully loaded in ZnO.

To illustrate the composition and chemical state of the elements in materials, XPS studies are performed. Figure 1a,b shows a full scan spectrum of the materials, and the results indicated that Zn and O are found to be the major elements in all the materials. From the Figure 1b spectrum, the photoelectron peak of Ni at 856 eV indicating that the Ni is successfully supported on ZnO. Figure 1c displays the orbital state of Zn 2p corresponding to the two asymmetric peaks of Zn 2p 1/2 and Zn 2p 3/2, respectively [32,33,34]. The fitting peaks located at binding energy of 1044.3 and 1021.2 eV are ascribed to Zn^2+^ in the ZnO and 5% NiO/ZnO, and other fitting peaks located at binding energy of 1045.3 and 1022.17 eV are attributed to Zn^2+^ in the MOF-5 and 5% Ni@MOF-5 [35]. Figure 1d shows the XPS spectrum of Ni2p. For 5% Ni@MOF-5 Ni 2p spectra (Figure 1d), the binding energy at 856.11 and 873.8 eV are attributed to Ni 2p 3/2 and Ni 2p 1/2 respectively, corresponding to Ni^2+^ in the sample. Characteristic XPS peaks for Zn 2p of 5% NiO/ZnO can be confirmed in Figure 1d at 872.43 and 855/853.1, which are attributed to the Ni 2p 1/2 and Ni 2p 3/2 states of the Ni2p orbital, respectively. There are two satellite peaks each corresponding to 2p 1/2 and 2p 3/2 confirming oxidation states of Ni [36]. The O 1s XPS spectra of materials are investigated as shown in Appendix A. It is generally that the peak at 530.2 eV belongs to the characteristic of O^2-^ of metal oxides, while the other peak at 531.25 eV is the typical of surface adsorbed oxygen.

The typical morphologies and microstructures of samples are shown in Figure 2. Figure 2a,b shows the SEM images of MOF-5 and ZnO materials, respectively. From Figure 2a, the size of the MOF-5 (< 2 μm) in this experiment is much smaller than that of the cube MOF-5 (20–200 μm) reported in the literature, indicating that the size of the crystal can be reduced under the condition of magnetic stirring [27]. As shown in the SEM of Figure 2b, the derived ZnO nanostructures retain the well-defined original morphology after the thermal decomposition of cube MOF-5 particles. During the thermal decomposition of the MOF-5 precursor into ZnO material, no significant dimensional shrinkage is observed, but a mass of holes appeared on the surface of the material, thus forming a porous ZnO structure [37]. The SEM images of 5% Ni@MOF-5 and 5% NiO/ZnO are given in Figure 2c,d, respectively. The SEM image of 5% Ni@MOF-5 composite is not significantly different from the morphology of MOF-5, indicating that the addition of Ni^2+^ do not destroy the morphology of the original crystal (Figure 2c). Figure 2d shows the morphology of 5% NiO/ZnO composites. Although 5% NiO/ZnO retains the original morphology of 5% Ni@MOF-5, its microstructure is quite different. After careful observation of Figure 2d, it is found that the surface of the composite becomes rough and there are many pores. The appearance of a mass of pores makes the material have higher surface area and rich active sites, which is conducive to gas diffusion and improves gas sensing properties [38]. The morphology of 25% and 50% NiO /ZnO composites are similar as 5% NiO /ZnO (Appendix A). Although 25% and 50% NiO/ZnO composites still maintain the original morphology of MOF-5, their morphology is not as regular as 5% NiO/ZnO. The surface of their materials has many cracks and the surface becomes extremely rough. The EDS element mapping was used to verify the compositions of 5% NiO/ZnO microstructures as Appendix A. As shown in Appendix A, Zn, Ni, and O elements are detected respectively, which testifies the existence of Ni element in the prepared composites. In addition, the good dispersion of Zn, O, and Ni elements indicates that Zn and Ni elements are uniformly dispersed in the composites, which is beneficial to enhance the gas sensing performances of the materials [31]. We tested the BET surface areas of pure ZnO and different ratio of NiO/ZnO samples. The results are 33.025 m^2^ g^−1^ (ZnO), 31.891 m^2^ g^−1^ (5% NiO/ZnO), 13.986 m^2^ g^−1^ (25% NiO/ZnO), 6.215 m^2^ g^−1^ (50% NiO/ZnO), respectively. As can be seen, the BET surface area of these samples decreases with the increase of Ni ions content, which may be caused by Ni elements entering the original structure and partially replacing the Zn atoms in the Zn_4_O cluster [27]. In addition, the increase of Ni content may clog the holes in the material and reduce the BET.

### 3.2. Gas Sensing Performance

Normally, temperature is one of the principal elements affecting the semiconductor gas sensor. Therefore, the following four materials (ZnO, 5% NiO/ZnO, 25% NiO/ZnO, and 50% NiO/ZnO) are tested with 500 ppm acetone as a function of the operating temperatures to determine the optimum temperature, as shown in Figure 3a. It is obvious that the response characteristics of the four materials gradually increases to the maximum value and then decreases with the increasing of operating temperature. The response value-temperature variation trend of all sensors is basically the same, and the maximum response value appeared at 340 °C. Among them, 5% NiO/ZnO has higher response to acetone gas than other sensors and its response value is 7.3 times as that of pure ZnO. We also found that 25% NiO/ZnO seems to have a worse performance than 50% NiO/ZnO. The possible reasons are as follows: On the one hand, the higher Ni ions content in the sample can enhance the catalytic activity of the material and make it easier for oxygen molecules to be adsorbed on the surface of the material, thus improving the gas sensing effect of the material to acetone [39,40]. On the other hand, the higher Ni ions content in the sample can enhance the *p-n* heterojunction in the material, so as to effectively improve the sensitivity of the material to acetone [39,41]. However, the higher Ni ions content in the sample will block the pore of the material and reduce the BET, which leads to the decrease of the sensitivity of the material to acetone. In this case, although the BET of 25% NiO/ZnO is higher than 50% NiO/ZnO, its catalytic activity is not as strong as 50% NiO/ZnO, so the synergy of these factors makes the sensitivity of 25% NiO/ZnO to acetone worse than 50% NiO/ZnO. The difference of material response value to various gases is one of a significant index to measure the quality of gas sensors. Figure 3b examines the selectivity of 5% NiO/ZnO and tests the response value of materials to 500 ppm methanol, acetone, ethanol, formaldehyde, methylbenzene, and ammonia at the optimum temperature. Obviously, the response of composite based on 5% NiO/ZnO to acetone is significantly higher than other test gases, proving that the composite has a good selectivity to acetone. The acetone gas sensing characteristics of the Ni^2+^-doped composite prepared in this work and the materials reported in other literatures are shown in Table 1 [42,43,44,45,46]. In Table 1, the response of Ni-doped ZnO thin film [45] to 100 ppm (half of the acetone concentration used in this work, 200 ppm) is 32 at a lower temperature of 200 °C, but the relationship between sensor response and gas concentration are not discussed in the literature, so this data can only be used as reference data.

The gas sensitivity of the ZnO, 5% NiO/ZnO, 25% NiO/ZnO, and 50% NiO/ZnO materials is verified by their responses to acetone concentration-response values. Figure 4a shows that the response value of each sensor gradually increased as the concentration of acetone increases from 10 to 500 ppm. Ni-doped ZnO improves the gas sensitivity of the original ZnO to acetone and the response amplitudes for the 5% NiO/ZnO to acetone is evidently higher than 25% NiO/ZnO and 50% NiO/ZnO, indicating the positive effect of Ni-doped on the acetone sensitivity of ZnO. Figure 4b shows a more intuitive relationship between the gas concentration and the response value of sensor. The response value of the materials is linear with the rises of acetone concentration from 10 to 100 ppm. The response growth becomes slower when the acetone concentration is over 100 ppm, indicating that the sensor gradually tends to be saturation. The responses for ZnO, 5% NiO/ZnO, 25% NiO/ZnO, and 50% NiO/ZnO to 500 ppm acetone are 13.285, 97.134, 17.611, and 35.217, respectively. Figure 4c displays the linear relationship between the response of 5% NiO/ZnO and acetone concentration (10–100 ppm). The correlation coefficient R^2^ is 0.98447, which is close to 1 and the concentration is almost linear with the response. Figure 4d gives the details of the transient resistance curve for 5% NiO/ZnO sensor to acetone (200 ppm) at 340 °C. The resistance of the sensor decreased rapidly by injecting acetone, and then increased to its initial value after releasing acetone from the chamber. In addition, it can be observed that the response-recover times of 5% NiO/ZnO to acetone (200 ppm) are 24 and 133 s, respectively. The faster response-recover time and higher response value of the 5% NiO/ZnO sensor determined its good detection ability for acetone [31].

Two necessary parameters, reproducibility and long-term stability, are the key to the application of gas sensor in practice. To research the stability and reproducibility under the same conditions, cycle experiments were carried out on the 5% NiO/ZnO sensor. The reproducibility of 5% NiO/ZnO was tested to 200 ppm acetone at the optimal temperature (340 °C). The response of 5% NiO/ZnO has almost no change to 200 ppm acetone gas in three cycles test, indicating that it has a noticeable repeatability character to acetone (Appendix A). In the long-term stability test of ZnO to 200 ppm acetone (Appendix A), the response value remains stable for a long time (30 days), revealing that the sensor has good stability for the detection of acetone.

### 3.3. Gas Sensing Mechanism

For the resistance-controlled gas sensor, the gas sensing mechanism can be explained by the resistance value change of sensor under the action of air or target gas. When the pure ZnO gas sensor is exposed to air, oxygen molecules adsorbed on the material surface capture free electrons from the material to form chemisorption oxygen (O_2_^−^, O^−^, and O^2−^), thus changing the electron distribution in the material and forming a depletion layer [33,47]. When the sensor is exposed to the acetone atmosphere, the acetone molecule will react with the adsorbed oxygen to release the trapped electrons, so that the depletion layer of the material is weakened, the electrical conductivity increases, and the resistance decreases, thus producing a gas-sensitive response. The oxygen adsorption process can be debated by equations as follows in Equations (1)–(4).
O_2_ (gas) ↔ O_2_ (adsorb)(1)
O_2_ (gas) + e^−^ ↔ O_2_^−^ (adsorb)(2)
O_2_^−^ (gas) + e^−^ ↔ 2O^−^ (adsorb)(3)
O^−^ (gas) + e^−^ ↔ O^2−^ (adsorb)(4)

Nevertheless, compared with pure ZnO, the gas sensing mechanism of NiO/ZnO gas sensor is slightly different. In the air, since the Fermi level of ZnO is higher than NiO, electrons will flow from *n*-ZnO to *p*-NiO while holes will flow in opposite directions until equilibrium is reached, resulting in the form of *p*-*n* heterojunction (Figure 5a) [41]. The form of *p*-*n* heterojunction causes the resistance of NiO/ZnO material to be higher than that of pure ZnO [48]. As shown in Figure 5b, when the material is exposed to acetone gas, the adsorbed oxygen reacts with the acetone molecules to release the captured electrons into the conduction band, resulting in a reduction in resistance. Meanwhile, the acetone gas reacts with holes in p-NiO to release electrons and combine with them to reduce holes concentration [49,50]. The chemical reaction of acetone molecule with adsorbed O^2-^ is shown as follows in Equation (5):CH_3_COCH_3_ + 8O^2−^ ↔ 3CO_2_ + 3H_2_O + 16e^−^(5)

The decrease of the carrier concentration on both sides of the *p*-*n* heterojunction leads to a decrease in the barrier height of the depletion layer, thereby further reducing the NiO/ZnO gas sensor [31]. To sum up, compared with pure ZnO materials, the resistance of NiO/ZnO materials in air will be greatly increased, while the resistance will be further reduced in acetone gas, so as to enhance the gas sensing response.

## 4. Conclusions

In summary, the bimetallic semiconductors prepared by Ni-doped MOF-5 significantly improve the sensitivity to acetone. Among them, the MOF-5 doped with 5% Ni^2+^ has the best gas sensing property, and its response value to 500 ppm acetone is 97.134, which is about 7.5 times higher than that of undoped one. Meanwhile, the sensing mechanism of NiO/ZnO materials is studied for acetone performances. Due to its good selectivity and reproducibility, it has a potential application value in the detection of acetone.

## Figures and Tables

**Figure 1 nanomaterials-10-00386-f001:**
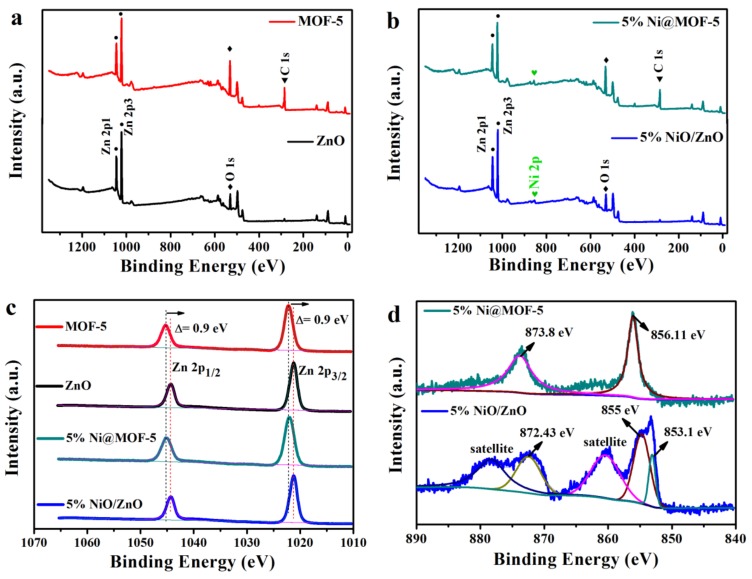
(**a**,**b**) Full scan spectrum of materials, (**c**) Zn 2p spectrum, and (**d**) Ni 2p spectrum.

**Figure 2 nanomaterials-10-00386-f002:**
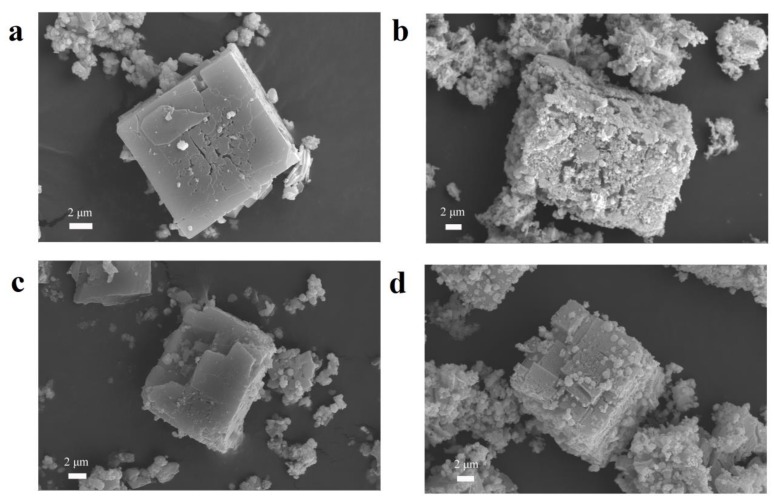
(**a**–**d**) SEM images of the metal oxide semiconductor (MOF)-5, ZnO, 5% Ni@MOF-5, and 5% NiO/ZnO microstructures.

**Figure 3 nanomaterials-10-00386-f003:**
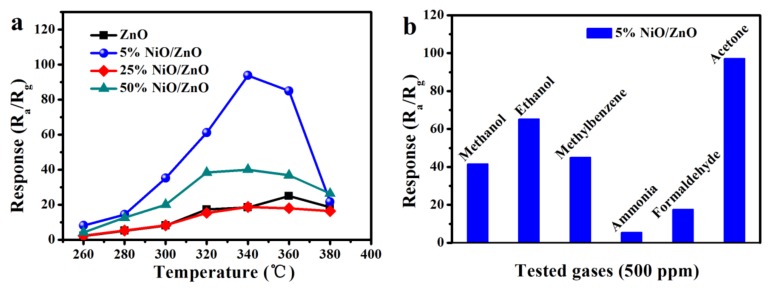
(**a**) Temperature-dependent responses of the samples towards 500 ppm acetone, (**b**) response of 5% NiO/ZnO to various gases at 340 °C.

**Figure 4 nanomaterials-10-00386-f004:**
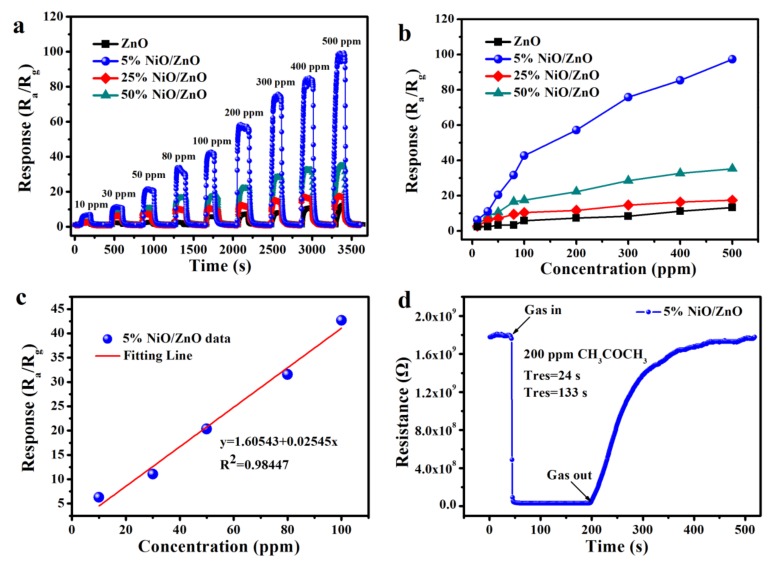
(**a**) Response-recovery curves of acetone concentration change at 340 °C, (**b**) linear relationship between concentration and response at 340 °C, (**c**) linear fitting curves in the range of 10–100 ppm, (**d**) transient resistance curve of 5% NiO/ZnO.

**Figure 5 nanomaterials-10-00386-f005:**
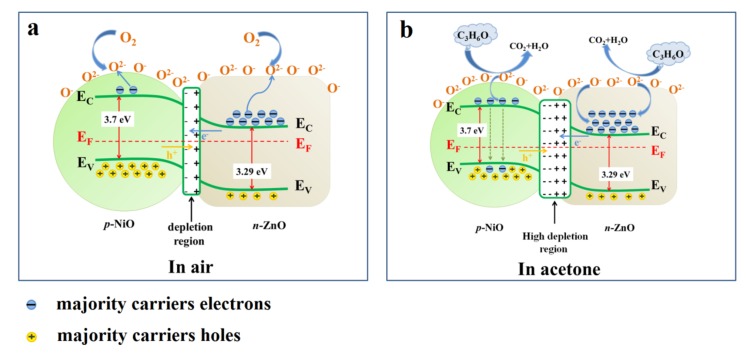
(**a**) Band diagram for ZnO/NiO interface in air, (**b**) band diagram for ZnO/NiO after interface in target gas.

**Table 1 nanomaterials-10-00386-t001:** Comparison of the gas sensing characteristics to acetone of various composites.

Materials	Acetone (ppm)	Temperature (°C)	Response	Res-Rev Time (s)	Ref.
3D hierarchically ZnO microsphere	200 ppm	330	25	11/17	[42]
ZnO nanotube	200 ppm	500	3.9	5/10	[43]
ZnO—(8 wt% NiO) Microflowers	200 ppm	300	37	3/41	[44]
Ni-doped ZnO thin film	100 ppm	200	32	-	[45]
Ni-doped ZnO plates	300 ppm	300	23	-	[46]
5% NiO/ZnO	200 ppm	340	58	24/133	this work

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
