# Peer review of "Enhanced Acetone Sensing Property of a Sacrificial Template Based on Cubic-Like MOF-5 Doped by Ni Nanoparticles"

_nanomaterials, 2020, doi:10.3390/nano10020386_

Round 1

Reviewer 1 Report

Please use the same tense throughout the paper. Some parts are written in past, some in present and some in the future tense (e.g. section 2.2, etc). Line 65: There is no verb in this sentence. Line 66: What part of the preparation is modified compared to [27-30] and what parts are the same other work? What are the contributions of the authors in designing new materials or methods? Please clarify to better state the novelty of the work. Section 3.2 and Fig 3: Did authors conduct any study to see if the operating temperature also affects the ratio of sensor sensitivities for different gases? I.e. if there is a temperature that the sensitivity toward acetone is highest compared to methanol or ethanol? Maximizing Sensitvity_Acetone/Sensitivity_Methanol? This can be further used to improve selectivity, however, at the cost of sensitivity. Or is it that the temperature increase from 260C to 340C would change the response (Ra/Rg) toward all those gases with the same level? This is not suggested to be conducted in this work if it hasn’t been done already. But the authors may consider it in future. Table 1: Ni doped ZnO thin film [42] has a response of 32 for 100ppm (half of the acetone concentration used in this work, 200ppm) at a much lower temperature, 200C compared to this work. Is the sensor response linear versus the gas concentration level? If yes, then [42] provides a better sensitivity at a much lower temperature. If the response is not linear, then it will add to the complexity of using the sensor for gas detection. Please clarify. Line 182: Looking at the significant figures of those numbers (digits, e.g. 13.285), are these responses calculated? What is the accuracy/error of these measurements? Line 198: Figure “S4a”?

Author Response

Response to Reviewer 1

We appreciate your positive comments and constructive suggestions. We have revised our manuscript based on your comments. The changes made are in yellow marked in the revised manuscript. The follow is our reply to your comments:

Point 1: Please use the same tense throughout the paper. Some parts are written in past, some in present and some in the future tense (e.g. section 2.2, etc). Line 65: There is no verb in this sentence.

Response 1: Thanks for your advice. The paper has been revised to the same tense according to your suggestion. And the changes made are in yellow marked in the revised manuscript. Line 65: We have added the verb in this sentence. The sentence is revised as follows:

 “The NiO/ZnO structure was synthesized by oil bath method.”

Point 2: Line 66: What part of the preparation is modified compared to [27-30] and what parts are the same other work? What are the contributions of the authors in designing new materials or methods? Please clarify to better state the novelty of the work.

Response 2: Thanks for your suggestion. In this study, Ni@MOF-5 was synthesized mainly by the method of oil bath stirring, and the required time of synthetic materials in this experiment was shortened to 8 hours compared with the literature reports. The experimental method is simple and convenient, and the size of MOF-5 obtained by magnetic stirring is smaller than that reported in literature. We have added a description of the size of the MOF-5 in the SEM section.

Point 3: Section 3.2 and Fig 3: Did authors conduct any study to see if the operating temperature also affects the ratio of sensor sensitivities for different gases? I.e. if there is a temperature that the sensitivity toward acetone is highest compared to methanol or ethanol? Maximizing Sensitvity_Acetone/Sensitivity_Methanol? This can be further used to improve selectivity, however, at the cost of sensitivity. Or is it that the temperature increase from 260C to 340C would change the response (Ra/Rg) toward all those gases with the same level? This is not suggested to be conducted in this work if it hasn’t been done already. But the authors may consider it in future.

Response 3: Thank you very much for your constructive suggestion. We tested the sensor's sensitivity to methanol, ethanol, methylbenzene, ammonia and formaldehyde at different temperatures. The results showed that the sensitivity of the sensor to these gases did not change significantly with the temperature change and no suitable temperature was found to determine the maximum sensitivity. In the end, we consider 340°C to be the best operating temperature.

Point 4: Table 1: Ni doped ZnO thin film [42] has a response of 32 for 100ppm (half of the acetone concentration used in this work, 200ppm) at a much lower temperature, 200C compared to this work. Is the sensor response linear versus the gas concentration level? If yes, then [42] provides a better sensitivity at a much lower temperature. If the response is not linear, then it will add to the complexity of using the sensor for gas detection. Please clarify.

Response 4: Thanks for your suggestion. Table 1: The relationship between sensor response and gas concentration is not discussed in literature [42], but its sensitivity to acetone gas is measured when exploring the selectivity of sensor, so this data can only be used as reference data. We have added an interpretation of the literature [42] in the manuscript.

Point 5: Line 182: Looking at the significant figures of those numbers (digits, e.g. 13.285), are these responses calculated? What is the accuracy/error of these measurements?

Response 5: Thanks for your suggestion. Line 182: These response values are obtained through the sensor test system and data acquisition system of CGS-4TPs (Beijing Elite Tech Co., Ltd., China), not through calculation. The measurement accuracy is within the range of ±0.5%.

Point 6: Line 198: Figure “S4a”?

Response 6: Thanks for your advice. Figure S4a: Repeatability measurement of the 5% NiO/ZnO to 200 ppm acetone at 340°C; Figure S4b: The long-term stability test of 5% NiO/ZnO to acetone (200 ppm).

Reviewer 2 Report

The Manuscript deals with the preparation, characterisation and application of Ni-doped MOF-5 based materials in the gas sensors field. The topic is surely current and interesting. Many experimental data are reported, and the adopted methodologies are appropriate. The article is clearly written in a concise way. While proposing results of interest, however it suffers from some minor + major criticisms that should be addressed before publication, as detailed hereinafter.

Authors report to have synthesised ZnO-based porous materials through MOF-5, that acts as template to get high specific areas. This claim appears to not be supported by experimental evidences. Authors should give data for specific area and pore volume for all the synthesised samples, together with the benchmark (ZnO). One would expect a decrease in specific area as long as the Ni load increases.

Authors report XPS and SEM results to explain dispersion and morphology of the samples. Nonetheless, similar analyses for samples loaded at 25% and 50% NiO miss. Please show them, or at least discuss about.

Figure 3a shows that the higher response (Ra/Rg) value is obtained for the samples at the lowest Ni content. This reflects this strict relationship between activity and dispersion, and morphology. Nonetheless, the sample at 25% seems to have a worst performance than the one at 50%. Please shed light on this. Same holds in Figure 4a.

Errors concerning the response measurements should be reported.

In Table 1 there is a comparison with data retrievable in literature. Nonetheless, there is no uniformity in the temperature conditions (450°C vs. 350°C). Please amend this.

Please specify the acronym OT in Table 1.

Author Response

Response to Reviewer 2

The Manuscript deals with the preparation, characterisation and application of Ni-doped MOF-5 based materials in the gas sensors field. The topic is surely current and interesting. Many experimental data are reported, and the adopted methodologies are appropriate. The article is clearly written in a concise way. While proposing results of interest, however it suffers from some minor + major criticisms that should be addressed before publication, as detailed hereinafter.

Response:

Thank you very much for your recognition of our work. We have revised our manuscript based on your comments. The changes made are in yellow marked in the revised manuscript. The follow is our reply to your comments:

Point 1: Authors report to have synthesised ZnO-based porous materials through MOF-5, that acts as template to get high specific areas. This claim appears to not be supported by experimental evidences. Authors should give data for specific area and pore volume for all the synthesised samples, together with the benchmark (ZnO). One would expect a decrease in specific area as long as the Ni load increases.

Response 1:Thanks for your suggestion. We have added the BET surface areas of pure ZnO and different ratio of NiO/ZnO samples. The results are 33.025 m2 g−1 (ZnO), 31.891 m2 g−1 (5% NiO/ZnO), 13.986 m2 g−1 (25% NiO/ZnO), 6.215 m2 g−1 (50% NiO/ZnO), respectively. As can be seen, the BET surface area of these samples decreases with the increase of Ni ions content, which may be caused by Ni elements entering the original structure and partially replacing the Zn atoms in the Zn4O cluster. In addition, the increase of Ni content may clog the holes in the material and reduce the BET.

Point 2: Authors report XPS and SEM results to explain dispersion and morphology of the samples. Nonetheless, similar analyses for samples loaded at 25% and 50% NiO miss. Please show them, or at least discuss about.

Response 2: Thank you for your constructive suggestion. Figure 1 is the SEM of the 25% NiO/ZnO (the left figure) and 50% NiO/ZnO (the right figure) that we tested previously. The figure shows that the morphology of 25% and 50% NiO/ZnO are not as regular as 5% NiO/ZnO, and there are many cracks on the surface of the material and the surface becomes extremely rough. Besides, the 25% and 50% NiO/ZnO are far less sensitive to acetone than 5% NiO/ZnO, so we didn't put it in the article. Based on your suggestion, we have added the SEM of the 25% and 50% NiO/ZnO (Figure S3 and Figure S4) in supplementary material. And, we have added discussion on 25% and 50% NiO/ZnO in the manuscript. The contents are as follows:

“The morphology of 25% and 50% NiO /ZnO composites are the similar as 5% NiO /ZnO. Although 25% and 50% NiO/ZnO composites still maintain the original morphology of MOF-5, their morphology is not as regular as 5% NiO/ZnO. The surface of their materials has many cracks and the surface becomes extremely rough.”

Point 3: Figure 3a shows that the higher response (Ra/Rg) value is obtained for the samples at the lowest Ni content. This reflects this strict relationship between activity and dispersion, and morphology. Nonetheless, the sample at 25% seems to have a worst performance than the one at 50%. Please shed light on this. Same holds in Figure 4a.

Response 3: Thank you very much for your constructive suggestion. The possible reasons for 25% NiO/ZnO performed worse than 50% NiO/ZnO are as follows:

On the one hand, the higher of Ni ions content in the sample can enhance the catalytic activity of the material and make it easier for oxygen molecules to be adsorbed on the surface of the material, thus improving the gas sensing effect of the material to acetone [1, 2]. On the other hand, the higher of Ni ions content in the sample can enhance the p-n heterojunction in the material, so as to effectively improve the sensitivity of the material to acetone [1, 3]. However, the higher Ni ions content in the sample will block the pore of the material and reduce the BET, which leads to the decrease of the sensitivity of the material to acetone. In this case, although the BET of 25% NiO/ZnO is higher than 50% NiO/ZnO, its catalytic activity is not as strong as 50% NiO/ZnO, so the synergy of these factors makes the sensitivity of 25% NiO/ZnO to acetone worse than 50% NiO/ZnO.

[1]  Zhang, Z.; Shao, C.; Li, X.; Wang, C.; Zhang, M.; Liu, Y. Electrospun nanofibers of p-type NiO/n-type ZnO heterojunctions with enhanced photocatalytic activity. ACS Appl. Mater. Inter. 2010, 2, 2915-2923.

[2]  Jang, D. M.; Kwak, I. H.; Kwon, E. L.; Jung, C. S.; Im, H. S.; Park, K.; Park, J. Transition-metal doping of oxide nanocrystals for enhanced catalytic oxygen evolution. J. Phys. Chem. C 2015, 119, 1921-1927.

[3]  Li, C.; Feng, C.; Qu, F.; Liu, J.; Zhu, L.; Lin, Y.; Wang, Y.; Li, F.; Zhou, J.; Ruan, S. Electrospun nanofibers of p-type NiO/n-type ZnO heterojunction with different NiO content and its influence on trimethylamine sensing properties. Sens. Actuators B chem. 2015, 207, 90-96.

Point 4: Errors concerning the response measurements should be reported.

Response 4: Thanks for your advice. The measurement accuracy is within the range of ±0.5% and the experiment is carried out under the condition of relative humidity (RH) of 20% ± 2%. The following is the analysis scope and accuracy of the sensor test system and monitoring system. Analyzing range: 1 Ω - 4 GΩ (precision: ±1 Ω, ±0.5% current value); temperature range: room temperature - 500 °C (precision: 1 °C); time limit: >1 s, continuously adjustable; monitoring system: chamber temperature (precision: 1 °C) and humidity (precision: 1%); evaporation: room temperature - 250 °C (precision: 1 °C).

Point 5: In Table 1 there is a comparison with data retrievable in literature. Nonetheless, there is no uniformity in the temperature conditions (450°C vs. 350°C). Please amend this.

Response 5: Thanks for your advice. Based on your suggestion, the temperature was revised [1]. And, the new reference was also added.

[1]   Ge, M.; Xuan, T.; Yin, G.; Lu, J.; He, D. Controllable synthesis of hierarchical-assembled porous ZnO microspheres for acetone gas sensor. Sens. Actuators B chem. 2015, 220, 356-361.

Point 6: Please specify the acronym OT in Table 1.

Response 6: Thank you for your constructive suggestion. The acronym OT stands for temperature and we have revised the acronym OT in Table 1.

Round 2

Reviewer 2 Report

The authors responded to all requests and appropriately modified the manuscript as requested by the reviewer.